# Looking Back, Going Forward: Understanding Cardiac Pathophysiology from Pressure–Volume Loops

**DOI:** 10.3390/biology13010055

**Published:** 2024-01-19

**Authors:** Ilaria Protti, Antoon van den Enden, Nicolas M. Van Mieghem, Christiaan L. Meuwese, Paolo Meani

**Affiliations:** 1Department of Intensive Care and Cardiology, Cardiovascular Institute, Thoraxcenter, Erasmus University Medical Center, 3012 Rotterdam, The Netherlands; i.protti@erasmusmc.nl (I.P.);; 2Department of Pathophysiology and Transplantation, Università degli Studi di Milano, 20122 Milan, Italy; 3Department of Cardiothoracic Surgery, Heart and Vascular Centre, Maastricht University Medical Centre, 6229 Maastricht, The Netherlands; 4Faculty of Health, Medicine and Life Sciences, Maastricht University, 6211 Maastricht, The Netherlands; 5Thoracic Research Center, Innovative Medical Forum, Collegium Medicum Nicolaus Copernicus University, 85-094 Bydgoszcz, Poland

**Keywords:** cardiac physiology, pressure–volume loops, left ventricle, right ventricle, ventriculo-arterial coupling

## Abstract

**Simple Summary:**

The cardiovascular pathophysiology’s complexity has been sharply rising, as a consequence of the advanced mechanical and pharmacological treatments. Pressure–volume loops (PVLs) depict in depth the intrinsic and extrinsic properties of both ventricles, as well as their interdependence. Therefore, the deep understanding of cardiac physiology based on PVLs provide a unique tool to interpret the increasingly demanding clinical scenarios.

**Abstract:**

Knowing cardiac physiology is essential for health care professionals working in the cardiovascular field. Pressure–volume loops (PVLs) offer a unique understanding of the myocardial working and have become pivotal in complex pathophysiological scenarios, such as profound cardiogenic shock or when mechanical circulatory supports are implemented. This review provides a comprehensive summary of the left and right ventricle physiology, based on the PVL interpretation.

## 1. Introduction

The use of mechanical circulatory support and multimodal pharmacological treatments has been sharply growing in the last decade [1]. As a consequence, a deep understanding of the cardiac physiology is a crucial requirement for health care professionals working in the cardiovascular field.

The heart is an electrically self-acting hydraulic pump consisting of two elastic muscle chambers connected in series, which simultaneously feed an approximately equal volume of blood into the pulmonary and systemic circulation. As first described by Suga and Sagawa [2], the force of the cardiac muscle strips at a given chamber volume generates a pressure that is related to the length of the muscle strips themselves. Since the force–length relationship of cardiac muscle underlies the ventricular pressure–volume relationship, a full understanding of cardiac pathophysiology is provided by ventricular pressure–volume loops (PVLs). The latter represent a unique approach for the real-time monitoring of both load–dependent and load–independent compounds of cardiac mechanics and, therefore, a complementary tool to figure out hemodynamic derangements and tailor appropriate treatments [3,4].

Cardiac physiology is presented below with reference to the left and right ventricular (LV and RV) PVLs.

## 2. Cardiac Cycle

By means of PVLs, each single cardiac cycle can be described by plotting instantaneous ventricular pressure on the y-axis and simultaneous ventricular volume on the x-axis, as shown in Figure 1A. At point *a*, ventricular pressure falls below atrial pressure, and the mitral valve opens, thus initiating ventricular filling. In a normally compliant heart, ventricular pressure is relatively low while ventricular volume is relatively high at the end of diastole (point *b*). Subsequently, isovolumic contraction begins. During this phase, the rise of wall tension results in an increased ventricular pressure as described by the Laplace law [5]: since this pressure is lower than the aortic pressure, the aortic valve remains closed, thus preventing changes in the ventricular volume. The rate of intraventricular pressure rise is an index of intrinsic ventricular contractility, due to its independence from the afterload. It can be described in terms of the maximum rate as dP/dt_max_, which is moderately sensitive to large increases in preload [6], or indexed for end-diastolic volume (EDV) to overcome preload dependency [7]. As soon as intraventricular pressure exceeds aortic pressure, the aortic valve opens and the ejection of blood into the arterial system occurs (point *c*). Following a peak chamber pressure, intraventricular pressure begins to fall until the end-systolic Pressure (ESP) point is reached (point *d*). Stroke volume (SV) is the amount of blood ejected by the ventricle with each cardiac cycle and equals the difference between end-diastolic and end-systolic volumes (SV = EDV − ESV). At the end of the ejection phase, the aortic valve closes and isovolumic relaxation begins. The rate of pressure decay defines ventricular lusitropy and is characterized by an exponential time constant of decay (τ), which reflects the rate of cross-bridge uncoupling within the myocytes [3]. Isovolumic relaxation continues until point *a*, when the cardiac cycle starts again.

[summary Table 1]

## 3. Intrinsic Properties

The loops are determined by the intrinsic properties of the heart: end–systolic pressure–volume relationship (ESPVR) and end–diastolic pressure–volume relationship (EDPVR). 

Within a physiological range, the *ESPVR* is an approximately straight line connecting each individual ESP points belonging to multiple PVL, experimentally obtained by changing both preload and afterload at constant ventricular contractility [8,9] (Figure 1A). This linear association can be expressed as follows [10]:ESP = Emax × (VES − V0)(1)
where ESP = end-systolic pressure, E_max_ = maximum ventricular elastance, V_ES_ = end-systolic volume, and V_0_ = volume axis intercept. 

The first factor of the aforementioned equation is E_max_ which is the slope of this linear association and can be expressed as follows: (2)$$\mathrm{Emax}\;=\frac{\Delta P}{\Delta V}\;[\frac{\mathrm{mmHg}}{\mathrm{mL}}].$$

Aortic ejection is prolonged shortly after E_max_ is reached, as a consequence of the aortic impedance and the blood inertia exiting the aortic valve. Therefore, E_max_ is not exactly reached at the end of systole but very close to it [5]. E_max_ represents an excellent index of intrinsic ventricular contractility, due to its load independence. Indeed, as shown by Suga and colleagues in canine left ventricles, under constant contractile state achieved through nerve section, extensive changes in heart rate and preload and afterload (or both) did not alter E_max_. However, the latter was profoundly changed by epinephrine infusion [11]. 

Another important factor is *V*_0_, which is considered as a functionally “*dead volume*” at which the ventricle cannot generate any force and, therefore, any pressure [10]. It is a function of cardiac size and can occur at negative volume values since ESPVR is approximated to be a linear relationship. However, the real ESPVR is always curvilinear, with a nonlinearity degree that depends on ventricular contractile state. This curvilinear fit (concave with respect to the volume axis) properly predicts the minute positive value of V_0_ [12].

A rise in contractility corresponds to an increased ventricular capacity to develop strength and, thus, pressure, for any given chamber volume. Nevertheless, the ventricular dead volume V_0_ does not change since the chamber size itself remains the same. As a consequence, in the pressure–volume diagram, the ESPVR can theoretically shift leftward or rightward (causing SV and ESP increases or drops, respectively) without any changes in V_0_ [9] (Figure 1B). However, V_0_ may also shift with changes in contractility. For instance, the infarcted ventricle shows E_max_ reduction accompanied by an increase in ventricular chamber size, which ultimately shifts the volume intercept [10,13]. Therefore, it is necessary to account for changes in both E_max_ and V_0_ to adequately interpret the ESPVR as a contractility index [4]. To mitigate this problem with the interpretation of V_0_, the ventricular volume at which the ESPVR reaches 120 mmHg (V_120_) is suggested as an alternative: the higher its value, the lower the ventricular contractility, and vice versa [4] (Figure 1B). 

The *EDPVR* is a curvilinear function enclosing the lower boundaries of multiple PVL obtained across different preloads and defines the passive diastolic properties of the ventricle [3] (Figure 1A). Nonlinearity accounts for the complexity of indexing diastolic properties. Therefore, in the past, a variety of exponential, cubic, and power curve fits have been applied to EDPVR, with the aim of developing a simple index of ventricular stiffness [7]. The latest proposed approach adapts the same curvilinear relationship to the following mathematically similar equation [5]: (3)$$P=S\times \log\;\left[\frac{Vm-V}{Vm-V0}\right],$$
where S = diastolic myocardial stiffness, a size-independent parameter having the units of stress; V_m_ = the maximum diastolic ventricular volume; V_0_ = volume axis intercept, the equilibrium diastolic volume at which pressure is zero. 

The *maximum volume* (*V_m_*) is asymptotically approached with the progressive increase in pressure, which would eventually lead to ventricular rupture [14]. V_m_ is not merely established by the dimension of the heart, but it is also strongly determined by the pericardium. The latter also tightly couples the right and left ventricular chambers and, therefore, their respective diastolic pressure, thus exerting a substantial effect on the left ventricular EDPVR. For this reason, the diastolic left ventricle should not be seen as “an unconstrained elastic shell of myocardium”, but rather as “a composite shell of stiff pericardium and elastic muscle, subdivided by the compliant interventricular septum” [15]. From a histological point of view, the curvilinear shape is due to different types of fibers being stretched at different pressure–volume ranges. Indeed, at a low pressure–volume range, the stretch of compliant elastin fibers determines only a small increase in pressure for a given volume increment. At larger volume ranges, on the other hand, the stretch of collagen fibers beyond their lengths allows the pressure to rise more steeply [7]. Therefore, the analysis of the EDPVR suggests the key diastolic features of the ventricle: at low volumes, it is highly compliant and fills easily, but becomes increasingly stiffer at higher volumes (Figure 1C). 

In clinical practice, the shifts of the EDPVR reflect changes in myocardial material properties (e.g., fibrosis and edema) and physiological or pathological remodeling [7]. A stiff ventricle shows a left-shifted EDPVR, whereas a compliant ventricle has a right-shifted EDPVR. Similarly, as described for systole, another index of diastolic properties is ventricular capacitance (V_30_), being defined as the volume at which ventricular pressure reaches 30 mmHg [4]. V_30_ reflects ventricular compliance and indexes the degree of remodeling (as in heart failure with reduced ejection fraction) or diastolic dysfunction (as in restrictive and hypertrophic cardiomyopathy) (Figure 1C). 

All the aforementioned intrinsic myocardial properties have been integrated by Suga and Sagawa into a unifying concept of time-varying elastance [16]. Indeed, it is possible to build a pressure–volume curve for each individual point in the cardiac cycle (Figure 1D). Since the slopes of all these lines have the dimension of elastance *(*∆*P*/∆*V*), the entire cardiac cycle can be seen in terms of cyclic variation in chamber elastance over time [5]. Thus, the time-varying elastance model consists of a single elastic element which cycles from low elastance (high compliance) during diastole to high elastance (low compliance) during systole. Therefore, this elastance increases gradually during the ejection phase, reaching its peak at the end of the systole [5,16].

[summary Table 2]

## 4. Extrinsic Properties

The position and shape of each single PVL are influenced by two fundamental extrinsic properties of the heart: preload and afterload. 

*Preload* is defined as the myocardial stretch at the end of diastole and is ultimately related to the maximal sarcomere length prior to systolic contraction [17]. It is indexed by end-diastolic pressure (EDP) or EDV [4], even though the relationship between filling pressure and filling volume varies in disease states involving diastolic dysfunction. According to the classic Frank–Starling law, the greater the stretch on the myocardium, the stronger the subsequent contraction [18]. This relationship is curvilinear, and therefore, at high preload, a further increase in volume would result in a smaller increase in cardiac output. Thereafter, many authors investigated the effects of preload changes on the PVL. In the experimental context, changes in preload can be obtained either in vivo, through inferior vena cava occlusion [3], or ex vivo. The ex vivo setting offers the unique possibility to investigate isolated changes in preload, afterload, and contractile state [8]. These experiments showed that, if afterload and contractility were kept constant, a decrease in preload would result in a smaller EDV and, consequently, a smaller and leftward shift of the PVL. Conversely, volume loading would lead to a larger and rightward shifted PVL, thus resulting in an increase in both ESP and SV (Figure 2A).

*Afterload* can be defined as the load the ventricle must overcome to eject the blood into the arterial system. It has been classically defined according to the modified Laplace law, assuming a spherical shape of the ventricle: (4)$$\tau =\frac{p.r}{h},\;[\mathrm{mmHg}]=[\frac{\left[\mathrm{mmHg}\right]\times \left[\mathrm{mm}\right]}{\left[\mathrm{mm}\right]}]$$
where τ = wall stress, p = intraventricular pressure, r = chamber radius, and h = wall thickness. 

However, this formula has two main limitations. First, ventricular shape is merely approximated and ignores the structure and geometry of the ventricle [17]. Second, the aortic vascular contribution is disregarded. As a consequence, the arterial input impedance was proposed as a viable alternative to estimate the afterload. It is derived from the Fournier analysis of aortic pressure and flow waves, and characterizes the mean and pulsatile properties of the vascular loading circuit [19]. However, the coupling of impedance to heart’s mechanical properties is mathematically complicated, due to its different domains (frequency and time domains, respectively). Therefore, Sunagawa et al. [20,21] developed the concept of “effective arterial elastance (E_a_)”, which best approximates the afterload based on the pressure–volume plane. The slope of the E_a_ line is approximately equal to the following [4]:(5)$$Ea=\frac{TPR}{T},\;[\frac{\mathrm{mmHg}}{\mathrm{ml}}]=[\frac{\mathrm{mmHg}\;x\;s}{\mathrm{mL}}\;x\;\frac{1}{s}]$$
where TPR = total peripheral resistance, and T = RR interval in seconds. 

However, at the end of ejection phase, the proximal artery contains an amount of blood volume which equals SV at a pressure that equals ESP. Therefore, E_a_ can be considered a lumped measure of the total arterial load that incorporates both resistive and pulsatile components, reflecting the net impact of this load on the ventricles [19]. Since E_a_ is graphically represented by the negative slope originating from ventricular elastance at ESP and intersecting the x axis at EDV (Figure 1A), it can be adequately estimated with the following simple formula [22]:(6)$$Ea=\frac{ESP}{SV},\;[\frac{\mathrm{mmHg}}{\mathrm{mL}}].$$

An increase in afterload causes an increase in the E_a_ slope along the ESPVR, and, consequently, a rightward shift of the PVL (if ESPVR remains constant), accompanied by a rise in ESP and a decrease in SV. In other words, an increased afterload results in an increased ESP at the expense of ejection, thus reducing the mechanical efficiency of the heart (Figure 2B).

[summary Table 3]

## 5. Myocardial Work and Oxygen Consumption

The time-varying elastance model provides the theoretical basis for quantifying the total mechanical energy produced with every single heartbeat. During the counterclockwise rotation of the instantaneous PVL, the ventricle generates an amount of mechanical energy to increase its elastance and allow ejection. This energy can be indexed by the area enclosed within the ESPVR, EDPVR, and the systolic limb of the same plot, named *pressure–volume area* (PVA) [23,24]. Only a part of the total energy produced by the ventricle is converted into external work and, thus, into cardiac output. Hence, the PVA equals the sum of two smaller areas, as assessed by the following formula:*PVA* = *SW* + *PE,*(7)
where SW = stroke work, the area within the loop, and PE = potential energy, the area bound by both the ESPVR and EDPVR and the isovolumic relaxation limb of the same plot [25] (Figure 3A).

*SW* is the external mechanical work that the ventricle performs to pump blood into the arterial system [24]. SW alone is not uniformly related to the myocardial oxygen consumption (MVO_2_) per beat, as it does not include the oxygen expenditure which occurs during isovolumic contraction and relaxation, when there is no ejection. Therefore, *PE* represents the potential energy accumulated during systole and not converted into external work, but rather stored at the end of the ejection phase in the elasticity of ventricular wall. In other words, it can be considered as the work against the viscoelastic properties of the myocardium [26].

As experimentally demonstrated by Suga and colleagues [27], it is not possible to convert all the PE into SW, thus reaching 100% of efficiency. However, as the relaxing ventricle can still perform the external mechanical work when allowed to reduce its volume against a decreased pressure load, up to 70% of this “wasted energy” can be effectively turned into SW. This work reasonably derives from some of the PE built during systole and then stored in ventricular wall. 

The MVO_2_ can be approximated by the means of coronary sinus blood flow (CS_bf_) and arterial–coronary sinus oxygen content difference [28]:

MVO_2_ = CS_bf_ × (CaO_2_ − C_cs_O_2_).(8)

In the perspective of PVL, a highly linear correlation was experimentally found between the MVO_2_ per beat and PVA, as energy expenditure is related to both internal and external work (Figure 3B). Therefore, the cardiac oxygen consumption can be expressed as follows [24,25]:(9)$${\mathrm{MVO}}_{2}=\left(a\;\times \;PVA\right)+b,$$
where *a* is the slope of this linear relationship and reflects the heart efficiency in converting chemical energy into mechanical energy, being close to 30% under physiological conditions [29,30], and *b* is the MVO_2_ axis intercept. Total MVO_2_ includes basal metabolism, intracellular calcium cycling (both independent of PVA), and cross-bridge cycling (directly proportional to PVA) [3]. As this straight line intercepts the MVO_2_ axis, even though the PVA work is theoretically reduced to zero, the heart keeps consuming oxygen to maintain basal metabolism and to recycle calcium back into the sarcoplasmic reticulum [5,30]. Therefore, for a given contractile state, these two parameters (*a* and *b*) remain constant [23]. In contrast, the MVO_2_/PVA relationship is affected by different inotropic states: an enhanced contractility results in an increased oxygen consumption intercept (*b*) without any changes in the slope (*a*). This illustrates that the extra energy consumption is due to the excitation–contraction coupling, whereas the basal metabolism remains unchanged [30]. On the contrary, chronotropism does not alter the same relationship [31]. 

It should be noted that the energy conversion efficiency (*a*), as assessed by MVO_2_/PVA relation’s slope, is different from the conventional mechanical efficiency of the heart (E), corresponding to the following [23]:(10)$$E=\frac{SW}{MVO2},\;[\frac{J}{\mathrm{mL}\;O2}].$$

Under physiological conditions, this ratio is approximately 25%. In other words, only about a quarter of oxygen consumed to produce energy is converted into external power [32]. The residual energy is dissipated as heat [33].

[summary Table 4]

## 6. Ventriculo-Arterial Coupling

The heart and the vascular system are anatomically and functionally linked. The interaction between ventricular elastance (E_es_) and arterial elastance (E_a_) during the ejection phase can be mathematically expressed as follows [34]:VAC = E_a_/E_es_.
where *VAC* defines the concept of ventriculo-arterial coupling, a major determinant of cardiac mechanoenergetics [35]. As mentioned above, E_es_ represents a load-independent myocardial contractility index. While E_a_ can be considered an integrative measure of the arterial system properties, including peripheral vascular resistances, the total arterial compliance, impedance, systolic and diastolic time intervals [35]. 

Consequently, VAC can be considered an appropriate tool to assess the cardiovascular performance, as it summarizes how preload, afterload, and contractility interact with each other to determine ventricular stroke volume and mean arterial pressure (MAP) [4]. SV and MAP can be expressed as follows [4,36]:(11)$$SV=\frac{EDV-V0}{1+\frac{Ea}{Ees}},$$
(12)$$MAP=0.9\;x\;\frac{\left(EDV-V0\right)}{\left(\frac{1}{Ees}+\frac{1}{Ea}\right)}.$$

Under physiological and resting conditions, cardiac function and arterial load are matched to maximize mechanical efficiency rather than the SW alone [37]. This is represented by a VAC ranging between 0.5 and 0.7 [3,36,37,38]. Under pathological or stressful conditions, the ventricle tends to maximize the SW at the expense of its efficiency, thus leading E_a_ and E_es_ to equalize (VAC~1) and yielding the best combination to deliver the maximal mechanical energy from the ventricle to the arterial tree [5,34,36]. As such, the heart works to minimize the oxygen consumption and maximize ventricular efficiency (metabolic optimization) during physiological conditions. However, under pathological conditions, maximizing the stroke work becomes a priority, thus leading to ventricular and arterial elastance equalization [37]. At very high VAC values (E_a_ > E_es_), the blood transfer from the ventricle to the arterial system becomes significantly inefficient, causing hemodynamic derangement and ineffective contraction (Figure 4) [35].

[summary Table 5]

## 7. Right Ventricle

The RV is a “thin-walled crescent-shaped structure coupled to systemic venous return on one side and to the pulmonary circulation on the other” [39]. From a physiological point of view, the primary role of the RV is to transfer all the venous return into the pulmonary circulation while preventing a rise in right atrial pressure [40]. This is possible because a healthy RV is coupled with a low-pressure and high-compliance arterial tree, thus resulting in lower wall stress during the entire cardiac cycle compared with the LV. 

Although from an anatomical point of view, the RV differs substantially from the LV, only minor differences in the classic PVL model exist, owing to its lower operating pressures and higher working volumes [41]. These differences make the appearance of a normal right ventricular PVL slightly different. As the RV pumps blood into a low-pressure system, the isovolumic contraction time is shortened, and thus, the ejection occurs early during a pressure rise. Consequently, peak chamber pressure is achieved quickly during systole. Then, the ESP decays nearly to the diastolic pressure, as the RV continues to eject blood into the pulmonary artery long after the beginning of its relaxation. All of this results in a more triangular round shape of a normal RV PV loop, compared to the square shape of normal LV PVL [42,43,44,45] (Figure 5). Therefore, despite the development of the same SV (since the average cardiac output must be roughly the same from both ventricles), RV external mechanical work, as indexed by the area within the triangular PVL, is proportionally smaller than that of square LV plot. The SW and MVO_2_ of the RV are approximately one-quarter of the LV energetic expenditure [45]. 

Since RV mechanoenergetic efficiency strictly depends on the low impedance imposed by the pulmonary circulation, its systolic function is highly sensitive to any acute changes in afterload. Indeed, an acute rise in pulmonary vascular resistances (as indexed by an increased E_a_ slope) promptly alters the RV PVL shape, which progressively becomes trapezoid or notched, with a late peak chamber pressure and an increased ESP (Figure 5) [44,46]. In case of compensatory hypertrophy, the RV develops a large contractile reserve when its afterload increases slowly [45]. A similar observation can be made for preload. Owing to the thinner wall and lower volume to wall surface area, healthy RV is a higher compliant chamber compared to the LV [39,42]. Indeed, the RV readily adapts to a chronically increased preload, being able to achieve even a threefold EDV with little or no change in EDP and, therefore, in basal compliance characteristics (rightward-shifted PVL with unchanged ESPVR and EDPVR [47]). On the contrary, the diastolic ventricular filling is markedly preload–dependent, and RV responds to an acute volume overload by dilating and altering its stiffness more than the LV (steeper EDPVR) [39,45].

Moreover, as for the left side, a contractile dysfunction responsible for RV failure manifests itself by lowering the ESPVR relationship slope (thus reducing E_max_), which in turn results in a marked reduction in both SV and ESP [47].

Lastly, VAC is a sensitive and independent estimate of cardiovascular system efficiency. The right VAC is more often calculated as the ratio between E_es_ and E_a_, the reversed ratio compared to the left ventricle [48]. As confirmed by prior studies, the optimal E_es_/E_a_ ratio ranges between 1.5–2.0 [49,50]. In the setting of chronically increased afterload, E_es_ initially increases due to the development of ventricular hypertrophy, in order to offset the increased E_a_ and, therefore, to maintain the RV–PA coupling. During an acute or chronic decompensated phase, a lack of additional contractile reserve leads to a disproportionate increase in E_a_ compared to E_es_, thus resulting in RV–PA decoupling, as indicated by a ratio below 0.6 to 0.8 and worse outcomes [48,49]. 

[summary Table 6]

## 8. Ventricular Interdependence

The concept of ventricular interdependence resides in an intimate anatomic relationship between the two ventricles, which includes the interlacing of superficial muscle bundles, a common interatrial and interventricular septum, and shared coronary blood flow and pericardium [41]. This interaction occurring during both systole and diastole and is amplified by the pericardium.

The *diastolic interaction* is exemplified by the fact that a distention from one of the two ventricles during the filling phase alters the compliance and geometry of the contralateral ventricle [51]. 

As originally demonstrated in postmortem-isolated hearts by Taylor and colleagues [52], changing pressure and volume in one ventricle profoundly influenced the contralateral size and compliance through the interventricular septum [53] and/or the pericardium [54]. Furthermore, in experimental setting, in the case of closed pericardium, RV pressure was the most powerful LV pressure determinant; on the contrary, as soon as the pericardium had been opened, LV geometry became predominant [15]. However, the ventricular interdependence becomes more intricate in in vivo animal, as well as in human physiology scenarios, as shown by Dell’Italia and colleagues [41,55]. First, pericardial restraint was the predominant mechanism when acute changes in preload and afterload affected both ventricular chambers simultaneously. Conversely, septum displacement and peripheral circulatory impedance appear dominant when acute changes affected only one ventricle. This mechanism represents an efficient cross-talking system between the two heart chambers. Indeed, during acute pulmonary embolism, the overloaded and distended RV shifted the interventricular septum leftward, thus decreasing LV chamber size and preload. At the same time, an acute reduction in LV stroke volume (due to the Starling mechanism) reduces systemic systolic pressure and, therefore, LV afterload. The simultaneous decrease in LV preload and afterload prevented the upward shift of its diastolic PVL, as differently showed in the previous experiments.

The *systolic ventricular interdependence* highlights the crucial role of the LV in generating right ventricular systolic pressure. Several authors in the past showed that up to 80% of right ventricular myocardium could be destroyed without any significant impact on right pump function [51,56,57]. Based on this finding, different theories were proposed varying from the central role of the LV in supporting the RV function to the need of a limited amount of RV fibers to ensure a normal RV performance [57]. The crucial role of the LV was further observed by experiments finding a significant impaired RV function as a consequence of the alterations in LV volume as well as LV free wall ischemia or a loss of its structural integrity [58,59]. Therefore, as stated by Damiano et al., the left ventricular contraction is fundamental for the right ventricular developed pressure and volume outflow, contributing over 50% of its mechanical work. 

[summary Table 7]

## 9. Future Perspective

Although PV loops represent the gold standard technique for thoroughly understanding cardiac pathophysiology, given the invasiveness of the procedure due to the cardiac catheterization; however, to date, they are still mostly relegated to the realm of research, with no real application in daily clinical practice yet. Therefore, the future is heading towards the development of computer simulation models [60,61], which allow the use of PV loops to be transferred from research laboratories to patients’ bedside, where they can effectively guide clinicians during the pathophysiology assessment and the hemodynamic management of patients admitted in the cardiovascular intensive care unit, especially if supported with mechanical circulatory supports.

## 10. Conclusions

Pressure–volume loops represent a unique tool to fully understand cardiac physiology. This results into a valuable means for clinicians to deeply appreciate the differences between right and left ventricular physiology, to understand how the ventricular chambers interact with each other or with implanted mechanical support systems and, accordingly, to prevent and adequately treat conditions in which physiology progresses towards pathology, resulting in hemodynamic derangements.

## Figures and Tables

**Figure 1 biology-13-00055-f001:**
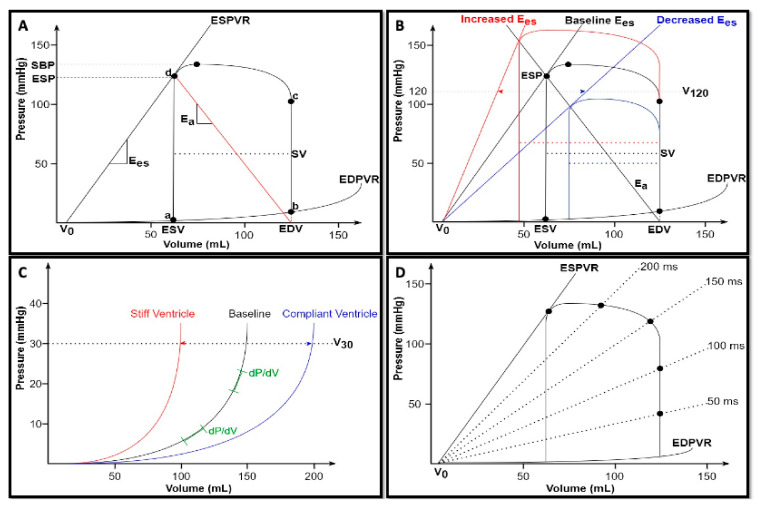
(**A**) Normal pressure–volume loop of the left ventricle. Ventricular volumes are on the *x*-axis, and ventricular pressures are on the *y*-axis. Each segment of the loop represents a phase of the cardiac cycle: *a–b,* ventricular filling; *b–c,* isovolumic contraction; *c–d,* ejection; and *d–a*, isovolumic relaxation. See text for details. (**B**) Changes in ventricular contractility. Under constant preload (EDV) and afterload (E_a_) conditions, a rise in contractility (red) results in a leftward shift of the ESPVR or V_120_, accompanied by an increase in both ESP and SV; on the contrary, a depression in myocardial contractility (blue) results in a rightward shift of the ESPVR or V_120_, with a consequent reduction in both ESP and SV. (**C**) Changes in diastolic properties. The curvilinear shape of the EDPVR suggests that the ventricle becomes increasingly stiffer at higher volumes (green). A stiff ventricle (red) shows a leftward shift of the EDPVR or V_30_, while a compliant ventricle shows (blue) a rightward shift of the EDPVR or V_30_. (**D**) Time-varying elastance. Pressure–volume curve of multiple points of the cardiac cycle which progressively increases the elastance from diastole to the end of systole (revised from [5]). ESV, end-systolic volume; EDV, end-diastolic volume; ESP, end-systolic pressure; SBP, systolic blood pressure; SV, stroke volume; ESPVR, end-systolic pressure-volume relationship; EDPVR, end-diastolic pressure-volume relationship; Ees, ventricular elastance; and Ea, arterial elastance.

**Figure 2 biology-13-00055-f002:**
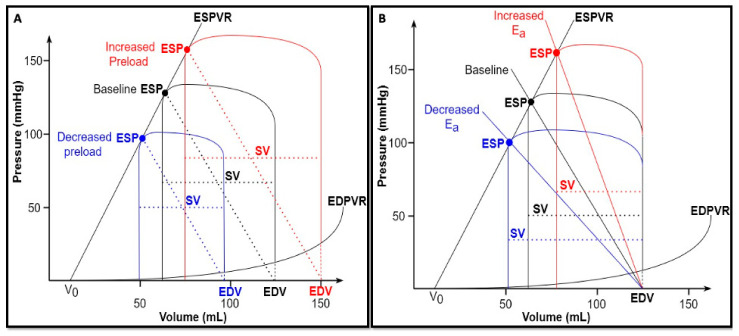
(**A**) Changes in preload. Under constant afterload (E_a_) and contractility (E_max_) conditions, a volume depletion (blue) results in a smaller and leftward-shifted PVL, with smaller EDV, ESP, and SV; on the contrary, a volume overload (red) results in a bigger and rightward-shifted PVL, with higher EDV, ESP, and SV. (**B**) Changes in afterload. Under constant preload (EDV) and contractility (E_es_) conditions, an increased afterload (rightward shift of E_a_—red) determines an increased ESP accompanied by a lower SV; on the contrary, a reduced afterload (leftward shift of E_a_—blue) determines a reduced ESP accompanied by a higher SV.

**Figure 3 biology-13-00055-f003:**
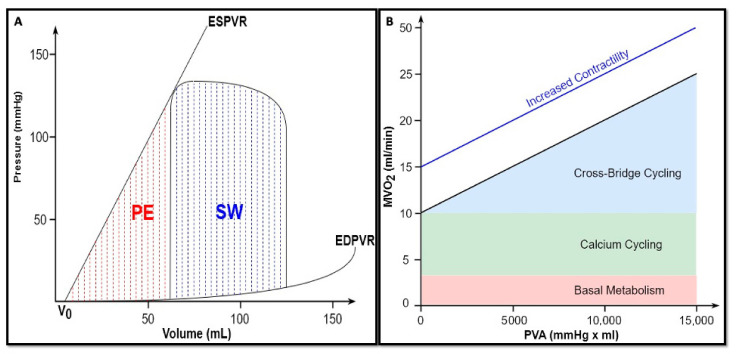
(**A**) Ventricular energetics. SW, stroke work, expressed by the following formula: $SW=\int_{EDV}^{ESV}P\left(V\right)dV-\int_{EDV}^{ESV}EDPVR\left(V\right)dV$. PE, potential energy, expressed by the following formula: $PE=\int_{V0}^{V1}ESPVRdV-\int_{V0}^{V1}EDPVR\left(V\right)dV$. (**B**) Myocardial oxygen consumption. The *x*-axis is the PVA. The *y*-axis is the MVO_2_. The association between MVO_2_ and PVA is highly linear and does not change in terms of slope and *y*-axis intercept under conditions of constant contractility. Conversely, an enhanced contractility increased the MVO_2_ intercept without any change in the slope, reflecting a higher oxygen consumption due to an additional cross-bridge cycling (revised from Bastos et al. [3]).

**Figure 4 biology-13-00055-f004:**
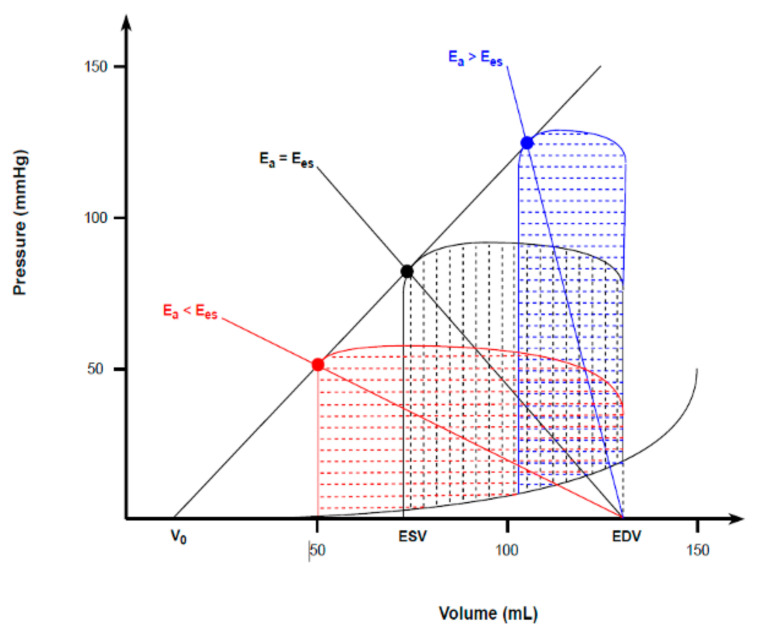
Ventriculo-arterial coupling and the corresponding ventricular energetics. Under resting conditions, the optimal VAC of < 1 (red) maximizes mechanical efficiency of the heart. Under pathological or stressful conditions, the ventricle maximizes the SW at the expense of mechanical efficiency, thus equalizing ventricular and arterial elastances (VAC = 1 − black). When VAC > 1 (blue), the extremely high PE makes ventricular contraction totally inefficient, causing hemodynamic derangements.

**Figure 5 biology-13-00055-f005:**
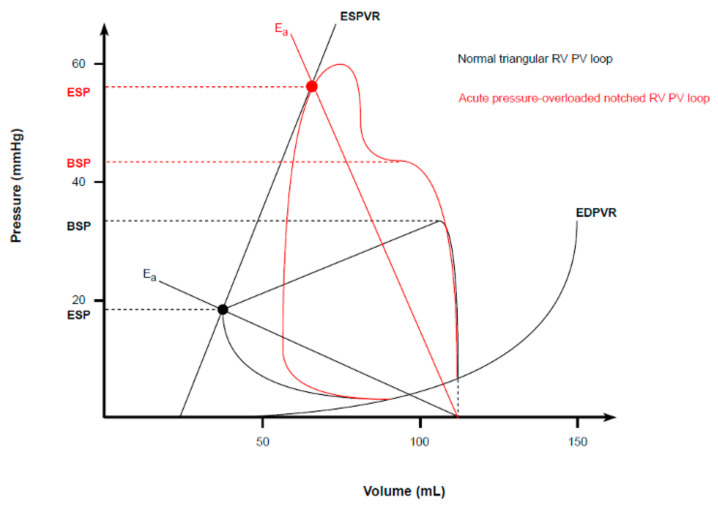
Normal and pathological RV PVL. Physiological RV PVL shows a triangular shape due to a shortened isovolumic contraction, an early peak chamber pressure, and a lower ESP point (closed to the diastolic pressure). An acute pressure overload alters the RV PVL shape into a trapezoid or notched one, with a late peak chamber pressure and an increased ESP.

**Table 1 biology-13-00055-t001:** Cardiac cycle.

(1) ventricular filling; (2) isovolumic contraction; (3) ejection; and (4) isovolumic relaxation.
The rate of intraventricular pressure rise (dP/dt_max_) and decay (τ) represent an index of intrinsic ventricular contractility and ventricular lusitropy, respectively.
The stroke volume equals the difference between end-diastolic and end-systolic volumes.

**Table 2 biology-13-00055-t002:** Intrinsic properties.

*E_max_* is the slope of the ESPVR and represents an excellent index of intrinsic ventricular contractility, due to its load independence.
A leftward shift in *ESPVR* reflects a rise in ventricular contractility; conversely, a rightward shift reflects a depression in ventricular contractility.
The *EDPVR* defines the passive diastolic properties of the ventricle, and its curvilinear shape suggests that the ventricle becomes increasingly stiffer at higher volumes.
The *EDPVR* is left-shifted in stiff ventricle, whereas it is right-shifted in compliant ventricle.
In the time-varying elastance model, the entire cardiac cycle can be observed in terms of cyclic variation in ventricular elastance over time.

**Table 3 biology-13-00055-t003:** Extrinsic properties.

Preload represents the maximal sarcomere stretch at end-diastole, and it is indexed by the *EDV*.
A decreased preload results in a smaller and leftward-shifted PVL, while a volume load leads to a larger and rightward-shifted PVL.
Afterload is optimally approximated by *E_a_*, a lumped measure of the total arterial load.
A rise in ventricular afterload causes an increase in the slope of *E_a_*, rightward shifting the PVL and resulting in increased ESP and decreased SV.

**Table 4 biology-13-00055-t004:** Myocardial work and oxygen consumption.

*PVA* indexes the total amount of mechanical energy produced by the ventricle at each cardiac cycle and equals the sum of two smaller areas, *SW* and *PE*.
*SW* represents the external mechanical work the ventricle performs to eject blood into the arterial system, whereas *PE* is the potential energy accumulated during systole and dissipated as heat.
The *PVA* correlates linearly with the *MVO*_2_, which includes basal metabolism, intracellular calcium cycling, and cross-bridge cycling.
*E* is indexed by the *SW/MVO*_2_ ratio and is approximately 25%.

**Table 5 biology-13-00055-t005:** VAC.

*VAC* is expressed by the ratio between ventricular and arterial elastance and represents a valuable tool to assess the performance of the cardiovascular system.
Under physiological and resting conditions, the *VAC* ranges between 0.5 to 0.7 to maximize *E* rather than *SW*.
Under pathological or stressful conditions, the ventricle tends to maximize *SW* at the expense of *E*, reaching a *VAC* of ~1.
When *E_a_* overcomes *E_es_* leading to a *VAC* of > 1, hemodynamic derangement occurs.

**Table 6 biology-13-00055-t006:** RV.

The triangular shape of RV PVL is explained by a shortened isovolumic contraction, an early peak chamber pressure, and an ESP close to the diastolic pressure, as ejection continues longer after relaxation begins.
RV accounts for approximately one-quarter of the LV energetic expenditure.
RV is highly sensitive to acute increases in afterload, which make the RV PVL shape trapezoid or notched, with a late peak chamber pressure and increased ESP.
The optimal RV VAC (E_es_/E_a_) ranges between 1.5 and 2 and becomes lower than 0.8 during an acute afterload increase, thus indicating RV–PA decoupling.

**Table 7 biology-13-00055-t007:** Ventricular interdependence.

In experimental settings, the interventricular septum plays a central role in determining the diastolic interaction between the two ventricles; in the case of a closed pericardium, the latter further enhances this diastolic interaction.
In human physiology, when acute preload and afterload changes simultaneously affect both ventricular chambers, the pericardium is the main determinant of diastolic interaction; conversely, when those acute changes affect only one ventricle, septum displacement and arterial impedance become predominant.
The systolic interdependence consists of LV contraction contribution to pressure and stroke volume development by the RV.

## Data Availability

Not applicable.

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
