# Peer review of "Looking Back, Going Forward: Understanding Cardiac Pathophysiology from Pressure–Volume Loops"

_biology, 2024, doi:10.3390/biology13010055_

Round 1

Reviewer 1 Report

Comments and Suggestions for Authors

Protti et al. reviewed the cardiac pressure-volume loops from a rather biophysics point of view with mathematics formulas. If the authors target medical doctors and cardiovascular researchers and no biophysicists as readership, the choice of journal is not the most appropriate, in my opinion. The title promises an understanding of the pathophysiological changes; however, this review mainly describes the physiology of the heart mechanics with short mentions of the underlying biochemical processes (e.g., the importance of the actin-myosin cross-bridge cycle in the Frank-Starling mechanism, positive inotropy and calcium ion overload, active and passive tension determinants in the sarcomeres, etc.).

1. In the present form, this manuscript seems to be a chapter of a physiology or biophysics textbook. The novelty is questionable: 9 of 59 references are no older than 5 years, and many are 20-30 years old. Please check the literature for novel articles and reviews, except for the original description of a law. Moreover, 59 references seem to be few to write a comprehensive review.

2. Modify the text according to the title. Add the explanation of the PV loop changes of common diseases (e.g., acute MI, acute HF, HFpEF, HFrEF, HFmrEF, acute and chronic cor pulmonale, etc.) under separate subheadings with corresponding figures.

3. Expand your MS with the underlying biochemical processes and important drug targets (e.g., positive inotrope drugs, myosin modulators, novel heart failure drugs, etc.) to give novelty of your review.

4. Increase the font size of the graphs. When printed, axis labels and other text are illegible in the Figures.

5. Expand your abstract after the modification of the MS.

Author Response

Dear reviewer,

Thank you for your appropriate comments and remarks.  

  1. In the present form, this manuscript seems to be a chapter of a physiology or biophysics textbook. The novelty is questionable: 9 of 59 references are no older than 5 years, and many are 20-30 years old. Please check the literature for novel articles and reviews, except for the original description of a law. Moreover, 59 references seem to be few to write a comprehensive review.

R: You are indeed right defining this manuscript a digression on cardiac physiology and biophysics, as that was exactly the purpose of this review. Since the Pressure – Volume Loops (PVL) offer a unique approach to deeply understand cardiac pathophysiology, we are extremely convinced of their usefulness from a clinical point of view, especially in the field of cardiovascular intensive care. Therefore we aimed to provide a proper tool for clinicians to develop a comprehensive understanding of cardiac pathophysiology and guarantee an optimal management of patients and their related mechanical circulatory support. Since the extensive understanding of PVL – based physiology represents the goal of this review, we described the cardiac physiology from a pure biophysics and mechanic point of view, trying to make it suitable even for clinicians without an in – depth mathematical – physical background. For the same reasons discussed above, it becomes clear why we based the review mostly on old references: since we are dealing with cardiac physiology interpreted according to PVL approach, all the concepts go back to about 20 – 30 years ago (as you well expressed). Therefore, the novelty of this comprehensive review, is to make a biomedical engineering topic usable for clinicians as well, so that it can be adopted not only in a research context but also in everyday clinical practice. We extremely believe that tailoring patient’s cardiovascular management on the understanding of PVL – driven pathophysiology can further optimize good clinical practice.

  1. Modify the text according to the title. Add the explanation of the PV loop changes of common diseases (e.g., acute MI, acute HF, HFpEF, HFrEF, HFmrEF, acute and chronic cor pulmonale, etc.) under separate subheadings with corresponding figures.

R: As discussed above, the aim of this first review is focused on the comprehensive understanding of pathophysiology by the means of PVL, therefore understanding the intrinsic and extrinsic properties of the myocardium and their respective variations, the myocardial oxygen consumption and the coupling with the arterial system. Hence, changes of PVL in the context of the main cardiovascular disease is not part our purpose. In this regard, we are indeed already working on a second review with a more clinical slant, in which would certainly be interesting to show such changes in PVL in specific clinical scenarios.

  1. Expand your MS with the underlying biochemical processes and important drug targets (e.g., positive inotrope drugs, myosin modulators, novel heart failure drugs, etc.) to give novelty of your review.

R: Although we believe this point is a very interesting suggestion, as anticipated earlier, since the topic  is narrowly focused on PVL and this approach drives a pure physical and mechanistic interpretation of cardiac pathophysiology, we decided to stick to the original proposed model, focusing only on this view of cardiac pathophysiology.

  1. Increase the font size of the graphs. When printed, axis labels and other text are illegible in the Figures.

R: The font size of the graphs has been accordingly changed.

  1. Expand your abstract after the modification of the MS.

R: For what has been said so far, the manuscript’s abstract is tailored on the review’s purpose as described above.

Reviewer 2 Report

Comments and Suggestions for Authors

Dear Authors,

The review manuscript titled "Looking Back, Going Forward: Understanding Cardiac Pathophysiology from Pressure-Volume Loops" provides a comprehensive review that explores various aspects of cardiac physiology and pathophysiology through the lens of Pressure-Volume Loops (PVL). The manuscript is very well written, maintaining a consistent flow throughout. Topics such as Pressure-Volume Loops (PVL), the cardiac cycle, ventricular mechanics, oxygen consumption, V-A coupling, and interdependence are well-covered.

While there are no major concerns, some minor issues need to be addressed:

1.     Since the topic of the review paper is “Looking Back, Going Forward …,” it would be beneficial to include an overview of the relevance and historical development of Pressure-Volume Loops in cardiac physiology. A brief discussion on how this technology has evolved and its current applications in clinical settings would be valuable.

2.     Could the authors provide a brief comparison between PVL and other diagnostic tools used in cardiac physiology? Including respective clinical examples would greatly enhance the paper.

3.     The manuscript does not address the possible integration of Artificial Intelligence (AI) and Machine Learning (ML) in understanding cardiac pathophysiology from Pressure-Volume Loops (PVL). I recommend the authors provide their thoughts on the incorporation of AI-enabled cardiac pathophysiology, diagnostic, and predictive capabilities related to PVL. You might consider citing this article [https://www.medrxiv.org/content/10.1101/2022.12.07.22283216v3.full-text], which offers insights into AI/ML utilization across various medical specialties. This reference will notably enhance the discussion section as it highlights how AI has made significant strides in medical use, with cardiovascular medicine being the second top medical specialty.

4.     Could there be a discussion on the limitations of PVL in different patient populations, such as those with congenital heart defects?

5.     A graphical abstract would also be helpful.

I believe my comments will help improve your review manuscript.

Best regards.

Comments on the Quality of English Language

Minor editing of English language required

Author Response

Dear reviewer,

Thank you for your appropriate comments and remarks. 

  1. Since the topic of the review paper is “Looking Back, Going Forward …,” it would be beneficial to include an overview of the relevance and historical development of Pressure-Volume Loops in cardiac physiology. A brief discussion on how this technology has evolved and its current applications in clinical settings would be valuable.

R: We have decided not to include a brief discussion on how the Pressure – Volume Loops (PVL) technology has been developed as the review in itself is already very long and detailed, so we wanted to avoid inserting further paragraphs that are not perfectly inherent to the topic. Instead, regarding the current applications of the PVL in clinical setting, we definitely agree that a short paragraph should be inserted in this regard (please see the amended text).

  1. Could the authors provide a brief comparison between PVL and other diagnostic tools used in cardiac physiology? Including respective clinical examples would greatly enhance the paper.

R: A comparison between the PVL and other diagnostic tools used to assess the cardiac physiology would be certainly of great interest. However this would probably be the subject of another review in its own right, outside the scope of the current review, which explicitly aims to provide clinicians with a detailed understanding of cardiac pathophysiology through the lens of PVL, so as to better tailor patient’s cardiovascular management and further optimize good clinical practice.

  1. The manuscript does not address the possible integration of Artificial Intelligence (AI) and Machine Learning (ML) in understanding cardiac pathophysiology from Pressure-Volume Loops (PVL). I recommend the authors provide their thoughts on the incorporation of AI-enabled cardiac pathophysiology, diagnostic, and predictive capabilities related to PVL. You might consider citing this article [https://www.medrxiv.org/content/10.1101/2022.12.07.22283216v3.full-text], which offers insights into AI/ML utilization across various medical specialties. This reference will notably enhance the discussion section as it highlights how AI has made significant strides in medical use, with cardiovascular medicine being the second top medical specialty.

R: As mentioned above, we agree that a short paragraph should be inserted in the field of “current and future applications of the PVL in clinical setting”, which necessarily involves new technologies such as the computer modelling of PVL, currently under study (please see the amended text).

  1. Could there be a discussion on the limitations of PVL in different patient populations, such as those with congenital heart defects?

R: The  PVLs are applicable in the context of any cardiac pathology as they provide a complex and detailed interpretation of all that comprises cardiovascular pathophysiology (intrinsic and extrinsic properties of the ventricle, myocardial oxygen consumption and ventriculo – arterial coupling). To our knowledge, they are also being studied in the setting of different congenital heart diseases.

  1. A graphical abstract would also be helpful.

R: Please see the Graphical Abstract, as requested. 

Reviewer 3 Report

Comments and Suggestions for Authors

The work is remarkable. I think it will contribute to the literature. However, could the conclusion part of the study be developed a little more?
32, 33, 34, 35, 36. paragraphs can we modify it as follows?
"The heart is an electrically self-acting hydraulic pump consisting of two elastic muscle chambers connected in series, which simultaneously feed an approximately equal volume of blood into the pulmonary and systemic circulation. As first described by Suga and Sagawa [2], the force of the cardiac muscle strips at a given chamber volume generates a pressure that is related to the length of the muscle strips themselves."

415.....the latter further amplifies enhances

Conclusion (Can we change this section as follows? We can also improve this section a little. Can we also mention the limitations of the research?)

"Pressure – Volume Loops represent a unique tool to fully understand cardiac physiology. This results into a valuable means for clinicians to deeply appreciate the differences between right and left ventricular physiology, to understand how the ventricular chambers interact with each other or with implanted mechanical support systems and, accordingly, to prevent and adequately treat conditions in which physiology progresses towards pathology, resulting in hemodynamic derangements."

Comments on the Quality of English Language

Minor editing of English language required

Author Response

Dear reviewer,

Thank you for your appropriate comments and remarks.

  1. 32, 33, 34, 35, 36. paragraphs can we modify it as follows? "The heart is an electrically self-acting hydraulic pump consisting of two elastic muscle chambers connected in series, which simultaneously feed an approximately equal volume of blood into the pulmonary and systemic circulation. As first described by Suga and Sagawa [2], the force of the cardiac muscle strips at a given chamber volume generates a pressure that is related to the length of the muscle strips themselves."

R: The sentences have been accordingly changed.

  1. ....the latter further amplifies enhances

R: The sentence has been accordingly changed.

  1. Conclusion (Can we change this section as follows? We can also improve this section a little. Can we also mention the limitations of the research?) "Pressure – Volume Loops represent a unique tool to fully understand cardiac physiology. This results into a valuable means for clinicians to deeply appreciate the differences between right and left ventricular physiology, to understand how the ventricular chambers interact with each other or with implanted mechanical support systems and, accordingly, to prevent and adequately treat conditions in which physiology progresses towards pathology, resulting in hemodynamic derangements."

R: The conclusions have been accordingly changed. For the limitations, please see the new paragraph “Limitations and Future Perspective in Clinical Scenario” in the amended text.

Reviewer 4 Report

Comments and Suggestions for Authors

The paper is very interesting and merits publication. No additionsl comment.

Author Response

Thanks for the reviewer's comments.

Round 2

Reviewer 1 Report

Comments and Suggestions for Authors

The authors answered my questions.

Reviewer 2 Report

Comments and Suggestions for Authors

The authors have provided responses to all comments.

Comments on the Quality of English Language

Minor editing of English language required.

Reviewer 3 Report

Comments and Suggestions for Authors

Thank you for the revisions. The article can be published in its current form.